# “Quo Vadis Diagnosis”: Application of Informatics in Early Detection of Pneumothorax

**DOI:** 10.3390/diagnostics13071305

**Published:** 2023-03-30

**Authors:** V. Dhilip Kumar, P. Rajesh, Oana Geman, Maria Daniela Craciun, Muhammad Arif, Roxana Filip

**Affiliations:** 1School of Computing, Vel Tech Rangarajan Dr. Sagunthala R&D Institute of Science and Technology, Chennai 600062, India; dhilipkumarit@gmail.com (V.D.K.); raji.maghudam08@gmail.com (P.R.); 2Department of Computers, Electronics and Automation, Faculty of Electrical Engineering and Computer Science, Stefan cel Mare University of Suceava, 720229 Suceava, Romania; 3Interdisciplinary Research Centre in Motricity Sciences and Human Health, Ştefan cel Mare University of Suceava, 720229 Suceava, Romania; 4Department of Computer Science, Superior University, Lahore 54000, Pakistan; muhammad.arif@cs.uol.edu.pk; 5Faculty of Medicine and Biological Sciences, Stefan cel Mare University of Suceava, 720229 Suceava, Romania; roxana.filip@usm.ro; 6Suceava Emergency County Hospital, 720224 Suceava, Romania

**Keywords:** deep learning techniques, pneumothorax, detection, X-ray images

## Abstract

A pneumothorax is a condition that occurs in the lung region when air enters the pleural space—the area between the lung and chest wall—causing the lung to collapse and making it difficult to breathe. This can happen spontaneously or as a result of an injury. The symptoms of a pneumothorax may include chest pain, shortness of breath, and rapid breathing. Although chest X-rays are commonly used to detect a pneumothorax, locating the affected area visually in X-ray images can be time-consuming and prone to errors. Existing computer technology for detecting this disease from X-rays is limited by three major issues, including class disparity, which causes overfitting, difficulty in detecting dark portions of the images, and vanishing gradient. To address these issues, we propose an ensemble deep learning model called PneumoNet, which uses synthetic images from data augmentation to address the class disparity issue and a segmentation system to identify dark areas. Finally, the issue of the vanishing gradient, which becomes very small during back propagation, can be addressed by hyperparameter optimization techniques that prevent the model from slowly converging and poorly performing. Our model achieved an accuracy of 98.41% on the Society for Imaging Informatics in Medicine pneumothorax dataset, outperforming other deep learning models and reducing the computation complexities in detecting the disease.

## 1. Introduction

A pneumothorax is a medical condition in which air or gas is present in the pleural space, the potential space between the lung and the chest wall. The presence of air or gas in this space can cause the lung to collapse, resulting in a reduced lung function and difficulty breathing. A pneumothorax can be classified based on the cause, such as spontaneous, traumatic, iatrogenic, or secondary. A spontaneous pneumothorax occurs without any obvious cause, typically in people with a history of lung disease or in those who smoke. It can happen when a small air pocket in the lung ruptures, causing air to leak into the pleural space. A traumatic pneumothorax is caused by an injury to the chest, an iatrogenic pneumothorax results from a medical procedure, such as a lung biopsy or the insertion of a chest tube, and a secondary pneumothorax occurs as a result of underlying lung diseases, such as emphysema, chronic obstructive pulmonary disease (COPD), or cystic fibrosis, which make the lung more susceptible to collapse. The diagnosis of a pneumothorax is typically made by physical examination and confirmed by radiologic imaging, such as chest X-rays or computed tomography (CT) scans.

Breathlessness and a sharp pain in the chest are the symptoms of a pneumothorax [1]. Prompt diagnosis of these symptoms is crucial because they may be dangerous in some circumstances. A serious pneumothorax can also result in dehydration, trauma, and perhaps even loss of life. Early detection is necessary for the rapid treatment of this illness. It is a common condition, with 35% prevalence and incidence in men, according to the authors of [2,3,4,5,6].

Song et al. [7] stated that while there are several medical methods available to detect pneumothoraces, such as magnetic resonance imaging, chest tomography scans, and ultrasound imaging, chest X-ray imaging is the most cost-effective option in terms of radiology. Chest X-rays are considered as the standard diagnostic tool for pneumothoraces due to their affordability and clear visual representation. In addition, chest X-rays are commonly used to diagnose and monitor various medical conditions, including tuberculosis, asthma, and disorders related to lung damage [8,9].

Pneumothorax detection is difficult because X-ray images have a low pixel resolution. If air is present even in low quantities in the laceration area (minor tear in the lung tissue due to injury), it becomes difficult to identify the disease. The assessment accuracy entirely depends on the doctors and medical practitioners involved. It is a challenging task to find skilled radiologists to review chest X-rays. Furthermore, each chest X-ray needs to be reviewed, and a report needs to be written by a qualified radiologist. As most radiologists are required to work long hours, fatigue-related misdiagnoses have increased. Hence, a computer-supported detection system is required to detect the disease on time with high accuracy to save human lives.

After the establishment of machine learning (ML) techniques, extensive research has been conducted in the medical sector on detecting hard-to-diagnose diseases within a short amount of time with the highest accuracy. The combined use of image processing, medical imaging, and computer technologies to identify health disorders in humans has led to significant breakthroughs. For example, the identification of melanoma [10], tachycardia, [11], and diabetic nephropathy [12] were among the first investigated using ML algorithms. Disease detection by chest radiographs is also eliciting considerable research attention. DL-based approaches for identifying lung nodules from X-ray images have been previously suggested by Huang et al. [13]. Through a 121-layered dense network to achieve an expert-level performance in detecting pneumonia and predicting numerous pulmonary illnesses by extracting the features from chest X-ray images, Zhu et al.’s work [14] demonstrated excellent performance in the detection of pneumonia using chest X-rays. 

Since pneumothorax identification involves finding tiny amounts of air stuck in between the chest and lungs, only a few research works have been conducted using chest X-rays. We propose to use chest X-ray images with profound DL algorithms to detect pneumothoraces and build a model by identifying several drawbacks in the existing approaches, which are detailed in the following. (1) Initially, the features of the chest X-ray images need to be extracted and should be classified to accurately detect pneumothoraces. The accuracy is fully dependent upon the quality of the noise free features extracted from the chest X-rays. The higher the quality of the input images, the higher the accuracy. The majority of existing approaches are inefficient at extracting the features from X-ray images without noise. (2) A lack of data augmentation results in a higher error rate. (3) Analyzing and extracting the features of X-ray images only in a single dimension also reduces the model’s accuracy.

This investigation was aimed towards the early diagnosis of a pneumothorax through X-rays by using a profound DL model with the highest accuracy by addressing the aforementioned drawbacks found in existing approaches. 

We proposed a DL model, PneumoNet, to classify X-ray images.We utilized a profound and publicly available dataset, the Society for Imaging Informatics in Medicine (SIIM) pneumothorax dataset from Kaggle [15], which is specifically for pneumothorax analyses.We proposed a channel optimization technique (COT) to improve the quality of the input image.We also used the affine image enhancement tool for image augmentation and noise removal.Our proposed PneumoNet model analyzed the input image in the obverse and flip sides of the image, thereby further improving the recognition rate.The results of our proposed approach were analyzed using various machine learning parameters to assess its accuracy. The results prove that our proposed approach outperforms the existing DL approaches in detecting pneumothoraces.

Section 2 contains the related work. The methods and materials of the proposed investigation are discussed in Section 3. The obtained results are analyzed in Section 4, and the work is discussed in Section 5 and concluded in Section 6.

## 2. Related Work

### 2.1. The Machine Learning Literature

Stein et al. [16] compared a deep CNN with a speed link to automatically detect and classify TB, which is similar to a pneumothorax, on the same two datasets. Nevertheless, the localization result produced using a feature map was not as impressive as the initially disclosed work, in which the greatest area under the curve (AUC) obtained was 0.93. Although all CNN models produced satisfactory results in the initial two trials for diagnosing pulmonary TB, the results of the following tests, which tackled the tricky but necessary task of detecting TB among an array of lung diseases, highlighted the difficultly of this task. 

In previous studies [17,18], pretrained CNN models were combined using the qualified majority to detect pneumothoraces. To create a method for detecting pneumothoraces by using chest X-rays images, Ruizhi et al. [19] tested conventional CNN architectures using open chest X-ray databases.

Lindsey et al. [20] used ultrasound images for detecting diseases. The authors of [21] used DL models and CNNs in the health sector to identify an application for which the models can be trained to recognize the given input. [22] used the support vector machine to train a model and segregated the input image into three dimensions to aid in identification of the affected region. 

### 2.2. The Deep Learning Literature

Th authors of [23] used a CNN technique based on pixel classification with a training set of around 120 chest X-rays to detect pneumothoraces. When assessing a test set of 96 chest X-rays, around 95% accuracy was achieved. To detect pneumothoraces, a surface interpretation method was integrated with the K-nearest neighbor algorithm. This suggested framework was tested on a dataset with 108 X-ray images, and the results indicated an 81% specificity and an 87% sensitivity. To regulate and extract the location of abnormalities in X-ray images using the U-Net framework and ResNet encoder, Sundaram et al. [24] used the SIIM–American College of Radiology (ACR) pneumothorax dataset with the AlexNet model to detect the disease. The initial step was to choose a symptoms that have a significant influence on diagnosis using recursive feature selection, and the algorithm then eliminates features that do not have as much impact as other factors. The second phase involves feeding the best diagnostic features to the AlexNet classification algorithm.

DL architectures such as AlexNet, GoogleNet, and ResNet were combined by Stefanus and Yaakob et al. [25] to categorize pneumothoraces. They tested different models and constructed the models using simple linear averaging, enabling it to produce around 92% accuracy. Dietterich et al. [26] used AlexNet and GoogleNet and discovered that a pretrained model is more precise. They used the composite index of the probability scores for every image to combine these models.

The researchers in [27] proposed computer-aided diagnostic approaches using a chest X-ray-based methodology to detect the initial condition of COVID-19 and compared it with other options (e.g., PCR, CT scan, etc.). They presented a heed DL model by using VGG-16. With the attention module, the spatial relationships of ROI are captured. The authors of [28] used the VGG-16 model’s fourth pooling layer and the attentive module to develop a novel DL technique for perfect classification. The results and evaluation demonstrated that the proposed method outperforms state-of-the-art methods with promising accuracy in detecting COVID-19 through X-ray images.

### 2.3. Heuristic and Hybrid Methods

Medical image segmentation was modeled using the heuristic red fox optimization algorithm by Jaszcz et al. [29]. Due to its heuristic characteristics, the red fox optimization algorithm automatically processes skin and retinal pictures. It models local and global movement using a random selection of Euclidean distance metrics. The suggested technique changes each pixel in a picture to black or white and monitors the color change and intensity. To evaluate the proposed method, an analysis of the thresholding values and their final value on X-ray images to generate a mask for segmentation was conducted. Each parameter variant had an accuracy of less than 80%. The segmentation effectiveness increased slightly from 20 to 50 people.

Kadry et al. [30] trained the InceptionV3 algorithm to identify pneumonia in chest X-ray images. Phases 1 through 3 of this technique are image collection, phase 2 is image enhancement via InceptionV3, phase 3 is feature compression using the Firefly algorithm, and phases 4 and 5 are multi-class classification and validation, respectively. 

The suggested method uses a four-class classifier and five-fold cross-validation to categories the X-ray into regular, moderate, medium, and extreme classes. The K-nearest neighbor (KNN) classifier’s experimental results showed that this method provided a classification accuracy of 85.18%.

Conventional feature extraction approaches have been used in early pneumothorax detection technologies. Edge detection of images has been used to determine the internal thoracic boundary in X-ray images [31], surface knowledge has been used to estimate portal hypertension cues [32], and the Cox transform [33] has been used to predict the occurrence of a pneumothorax in local intensity histograms. The diagnosis accuracy of such algorithms is currently relatively low, owing to the reddened appearance, which leads to the inability to distinguish between different human lungs and pneumothoraces.

Hamde [34] conceived and demonstrated machines that develop methods to assess pulmonary disease severity by using chest X-rays of congestive heart failure with edemas by utilizing a clinical -imaging dataset with around 30,000 patient images. From X-rays, Riasatian’s team [35] isolated the lung region based on the Gabor selection algorithm on an X-ray image by using anatomical techniques. The roughness and the form of the segmented dataset’s characteristics were computed.

Park et al. [36] proposed end-to-end pneumothorax classification, which analyzes the entire chest X-ray in a quick pass-through patch-based technique which applies a neural network recursively. This technique categorizes specific X-ray fragments to predict whether every fragment includes a pneumothorax. The authors concluded that although the proposed method produces acceptable accuracy, it cannot segregate specific spots in the image. Chest X-rays are extensively used and are inexpensive, enabling easy design of DL techniques. An overview of a few of the existing works is presented in Table 1.

## 3. Methods and Materials

We propose a DL method to identify the existence of a pneumothorax in X-ray images. Our proposed work utilized the features of ImageNet and ResNet pretrained models. Although the accuracies achieved by both models are around 93% and 95%, respectively, the feature maps obtained from both models suffer from vanishing gradient issues [38] owing to the back propagation of the output through the input layers. Any changes made in the input layer are not fully reflected in the output feature map that is back propagated to the input layer. For all propagations, the gradient value (change in the input layer) decreases stepwise and finally reaches zero. Accordingly, it does not reflect the change in the input layer. The vanishing gradient issue quickly occurs if the depth of the model is too large. It makes the training the model biased. If the model is made to be shallow to avoid the vanishing gradient issue, the network performance becomes too weak. Overall, this makes the model unable to identify dark regions in the input X-ray images. This major limitation is addressed by our proposed PneumoNet model. The residual part presented in the proposed model effectively reduces this problem. Figure 1 shows how the problem is addressed. When gradient reduction occurs in feature map 1, 2, and 3, the gradient value can be directly sent to feature map 1, thereby making the cross-layer propagation effective. 

Hence, regardless of the size of the gradient (change in input feature), the residual block in the proposed PneumoNet model retains the exact value of the weight, which is closer to the values changed in the input layer. Consequently, the gradient value becomes the same as the one present in the input feature map. The proposed architecture is shown in Figure 2.

### 3.1. Dataset

We used the SIIM-ACR dataset for our investigation. The medical health body for those interested in the present and potential applications of informatics in medical imaging is the SIIM. Their goal is to promote medical diagnostic informatics throughout the organization through scientific research, training, and creativity. The information present in the dataset comprises images formatted using the digital imaging and communications in medicine program with run-length-encoded masks and image identification as inscriptions. It is a globally used format to represent images in digital format, specifically medical images. Each image is encoded with a binary mask and stored in a separate folder. The meta information of every image contains the file location, the test person’s age, and the duration of disease, as well as the time of identifying the modality of the patient with their sex, date, and time of examination on a particular part of the body. 

Pneumothorax cases are visible in some images, and they are denoted by encapsulated digital overlays within annotations. Several annotations can be found on some training images. The dataset is open source and can be downloaded from Kaggle. The dataset contains around 3116 X-ray images of the chest in .png format of 1024 × 1024 image size labeled for training and testing the model. After data augmentation, around 9400 images are labeled for training, in which 8300 are pneumothorax-affected images and the remaining 1100 are unaffected. We used 655 original images (non-augmented images) to test the model, where 382 images were affected with pneumothorax and 273 samples were without pneumothorax. All the images were resized to 124 × 124 pixel size. The dataset is provided with run-length encode value, so it can be utilized for segmentation later.

A comma separated value file was created for all images present in the dataset. It is encoded with a mask value of −1 if the DICOM image does not contain a pneumothorax. If the image marks the presence of a pneumothorax, the pixels of the images are completely encoded and a null-valued mask is created for all images. This way of masking the image makes the segmentation process in line with the run-length encoding values. Table 2 lists the dataset details. Samples of chest X-ray images from the dataset are shown in Figure 3. 

### 3.2. Data Preprocessing

The original size of the image was 1024 × 1024, which was resized to the pixel resolution of 124 × 124. The benefit of this reduced size is that only a small amount of model parameters are required and the model can be trained faster. We incorporated the bicubic interpolation method to verify the importance of the resized image. The affine enhancement tool was used for image augmentation. Using the affine tool is powerful for OpenCV as the framework of this tool is simple and can be adapted for image segmentation, classification, and augmentation. It also supported noisy image fine tuning by converting it pixel by pixel.

### 3.3. Data Augmentation

Given that the performance increases with increased size of dataset, data augmentation plays crucial role in image classification. 

Data augmentation is a method for increasing the number of data items in a dataset used to train a model. The technique augments them using fundamental image processing techniques such as panning, spinning, scaling, and padding. These altered images derived from the existing image collection are then added to the dataset, thereby increasing the size of the dataset used to train the model.

In addition to expanding the dataset, this method provided the learning model with additional learning features. This study utilized two image processing operations, spinning and inversion, to augment the data. During the first data enhancement phase, 120 X-ray images were inverted to produce an additional 120 images. This operation resulted in an increase of 240 images in the resulting dataset. During the second phase, a 90° rotation was applied to the original 120 images to produce 120 additional images, then by 180° to again produce 120 additional images, and finally by 270° to produce a further 120 additional images. Similarly, images were spun, resulting in the same amount of additional images. The output of these operations was a dataset comprising 720 chest X-ray images. The process was continued until the dataset contained 3000 image samples. The image quality was further increased by resetting some parameters as follows: (1) the brightness was augmented to a max factor of 0.1, (2) the contrast augmented by rescaling the original image to 5%, (3) the gamma factor was augmented to 125, and (4) the contrast was augmented to 0.1. Some samples of the augmented images are shown in Figure 4. Table 3 contains the augmented data of the chest X-ray images.

One augmented image from every folder was selected to verify the correctness of the augmented image. A Gaussian noise variance with the range of 15 to 35, a maximum standard deviation with 0.5 probability, and a kernel size of 5 were the parameters used to verify the correctness of the augmented image. All augmented images were flipped 20° horizontally, and 0.02 is fixed as the relative distance for image shifting. Then, the augmented image was input into the proposed classifier during training. If the resultant response was positive, other images in the augmented folder would be completely accepted for training the model further.

### 3.4. Image Segmentation

The proposed PneumoNet architecture consists of a contracting path and an expanding path, which allows for the efficient capture of contextual information and the learning of complex features from the X-ray images. The contracting path of the PneumoNet architecture consists of convolutional and max pooling layers that reduce the spatial resolution of the input image, while increasing the number of feature channels. This path captures the context of the image and learns the features that are important for segmentation. The expanding path of the PneumoNet architecture consists of deconvolutional and up-sampling layers that increase the spatial resolution of the output image while decreasing the number of feature channels. This path uses the features learned in the contracting path to generate an accurate segmentation map. This ensures the model can accurately classify the dark portions present in the input chest X-ray images performed at different convolutions. This is achieved through the augmented data resulting from data augmentation techniques such as panning, spinning, scaling, and padding. The classification results obtained by the proposed work are better than the existing approaches that did not use any segmentation methods. This ensures the proposed architecture performs accurate segmentation with limited training data and makes it a useful tool for pneumothorax detection, where annotated data may be limited.

### 3.5. Proposed PneumoNet Model

Every channel presented in our proposed PneumoNet is sufficiently deep, with a huge number of feature maps such that it is capable of recognizing every feature of pneumothoraces with different patterns. We added COT to further improve our proposed network. COT reduced the loss occurring at every channel owing to the change in input feature map during convolution. By simulating the dependability of every channel and adapting the features one by one through all channels, the COT block enhances the network’s pneumothorax identification abilities. Through this process, the network can be trained to discriminately improve the features that contain valuable data and subdue those that do not use relevant data related to pneumothorax image classification. A structural representation of COT is shown in Figure 5.

Equations (1)–(4) express how the COT improves network performance. In these equations, ‘*G*’ is the input to the model; ‘*ri*’ and ‘*ti*’ are the values before and after optimizing the channel, respectively; ‘*wt*’ are the weights of the input; ‘*m*’ stores the average of weights; ‘*v*’ and ‘*w*’ are the activation function value and average pooling value, respectively, and ‘*k*’ is the counting value for every input ‘*i*,*j*’.
(1)$$G=f\left(r_{i\;,}t_{i}\right)$$
(2)$$t_{i}=f\left(m,w_{t}\right)$$
(3)$$\;m=f\left(r_{i}\right)=\frac{1}{v\times w\;}\sum_{i=1}^{v}\;\sum_{j=1}^{w}r\left(i,j\right)$$
(4)$$r_{i}=\sum_{k=1}^{s}v_{t\;\;\;}^{r}\times m$$

The information related to the characteristics ‘*G*’ of the input is represented as 1 × 1 × *s* from *v × w* owing to the pooling operation. In the first fully connected layer, the value of ‘*r*’ is stored as 16 in ‘*G*’. In the second layer of the fully connected layer, ‘*G*’ changes to 1 × 1 × *s.* The convolution output is obtained from ‘*r_i_ (i, j)*’.

In the first step, an explanatory channel is formed by collecting all spatial characteristics of each and every channel present the model. Second, based on the dependency of every channel, the feature map ‘*f_n_*’ is fine tuned. This fine-tuned feature map is the output of the PneumoNet model. The advantages of the COT have effects on the entire network. The image is flattened using a flattening layer. 

The classification of the input X-ray image is prepared using the fully connected layer. Ultimately, the results are obtained through Equation (5), which is the SoftMax function, ‘*S*’:(5)$$s=\frac{f_{n}}{\sum_{i=0}^{n}f_{i}}\;$$

Apart from the COT, PneumoNet is built with an encode and decode pack. The encode pack supports the extraction of more features from the images obtained from the COT. In other words, the image is fine-tuned again for accurate classification. This downscaling method is conducted using the pooling layers with a stride size of less than one. The decode pack of the model performs the up-scaling process for the input image. The generation and extraction of features are performed using the encoding and decoding pack of the model. Convergence of the input image is achieved by collecting all convolution layers through batch normalization. Our model uses a chest X-ray image as input and classifies the image as pneumothorax affected or unaffected. Binary cross-entropy serves as the loss function (Equation (6)). In some cases, asymmetric values,$\;\left(\bar{x_{i}}\right)$ of positive and negative instances, $x_{i}$, can be reduced using the loss function, and the estimated gradient additionally becomes more stable for initial value ‘*i*’ to end input ‘*n*’. All three branches used binary cross-entropy as their loss function, but their weighting indices are unique.
(6)$$Loss\_func=-\sum_{i=0}^{n}\left[x_{i}\times \log\left(\bar{x_{l}}\right)+\left(1-x_{i}\right)\log\left(1-\bar{x_{i}}\right)\right]$$

A step-by-step process is performed to classify the pneumothorax-affected or -unaffected images. For the model to view the X-ray image at all angles and to perform accurate labeling, all augmented input images underwent various geometrical transformations, G. Our proposed approach stores the cumulative weights of the best network and memorizes it every time the model reaches the local optimum, L. The average of L is the overall classification accuracy of the model.

We conducted a five-fold cross-validation for this investigation. The data were equally separated into five distinct folds. A data point is selected in such a way that no data crosses the other fold’s data point. The ratio for training and testing is 8:2. The accuracy is calculated as the average of all five-fold results.

### 3.6. Hyper Parameter Optimization

Tensor flow and Keras were used to investigate with Python using NVIDIA, 3080RTX, Ti. We used the Adam optimizer with a learning rate of 0.0001 and a batch size of 64 for 60 epochs. Grid search was utilized to fine tune the model. The model performance was optimized by verifying the output for the continuous training of five epochs. If the performance was not improved, the learning rate was condensed to 1/10 of its initial value. To prevent our model from overfitting, training was stopped if the performance was not improved after 15 epochs. 

We found that the maximum validation accuracy reached its highest point after employing 64 batches, and then began to decline on either side. This is because at a particular batch size, the capability of the proposed model to extrapolate starts to deteriorate; the accuracy rate was 96.82% for a batch size of 128 and 91.02% for a batch size of 256. When the batch size was fixed at 64, the accuracy was 98.81%, and thus it was decided that the batch size should be 64.

We began our search for the best learning rate using the PneumoNet model and 64 as our batch size, rather than assuming that 0.0004 was the correct value. The accuracy rate reached its highest point at 0.00017, and subsequent values were either higher or lower. If the learning rate is set too low, the model will take more time to arrive at the best solution, and it is possible that it will even misinterpret the data. On the contrary, if the learning rate is too high, the model will not be able to converge because the strides are too big. With only a learning rate of 0.0001, we were able to reach the best possible validation accuracy of 98.81%.

Dropout is a method for improving accuracy and avoiding overfitting in DL models by arbitrarily removing units from layers in the training process. This forces a neural network to learn with less data, which improves its generalization abilities. The optimal solution using the proposed PneumoNet model resulted in a validation accuracy of 98.41%, with a dropout rate of 10.59%, a learning rate of 0.0001, and a batch size of 64. The obtained results are shown in Table 4.

### 3.7. Computational Complexities

The computational complexities of the proposed approach are as follows. The inference time for the proposed approach is 0.381 ms and the model size is 32.11 KB. The proposed PneumoNet model outperformed Efficient Net and MobileNet in terms of accuracy while using considerably less computational complexity (12.5 G FLOPS) and fewer parameters (7.61 million) than both of these networks. Table 5 contains the computational complexity details of our approach.

## 4. Results

The results produced by our proposed PneumoNet model are presented and discussed in this section. Figure 6 shows the outcome of the proposed classifier when it has identified the disease.

The performance of the proposed model was analyzed using a few DL parameters. We used *Tr. Pt* (True Positive), *Tr. Nt* (True Negative), *Fs. Nt* (False Negative), and *Fs. Pt* (False Positive) metrics to classify the predicted images.

*Tr. Pt* (True Positive): This group contains around 9817 correctly classified pneumothorax images, which contains 99% air pockets in the lungs (affected).

*Tr. Nt* (True Negative): This group contains 26 images classified as having air pockets in the lungs but in reality they do not have air pockets (unaffected).

*Fs. Nt* (False Negative): This group contains chest X-ray images that are incorrectly classified as having air pockets (affected).

*Fs. Pt* (False Positive): This group contains chest X-ray images that are correctly classified as not having air pockets (unaffected).

Equations (7)–(10) are used to calculate the model performance.
(7)$$\mathrm{Accuracy}=\frac{Tr.Nt+Tr.Pt}{Tr.Nt+Fs.Pt+Fs.Nt+Tr.Pt}$$
(8)$$\;\mathrm{Recall}=\frac{Tr.Pt}{Fs.Nt+Tr.Pt}$$
(9)$$\mathrm{Specificity}=\frac{Tr.Nt}{Fs.Pt+Tr.Nt}$$
(10)$$\;\mathrm{Precision}=\frac{Tr.Pt}{Fs.Pt+Tr.Pt}$$

Figure 7 presents the numeric details of the classification results in the form of a confusion matrix. The matrix expresses the total count of X-ray images that are classified as pneumothorax affected and unaffected.

The classification results obtained by the proposed model are tabulated in Table 6. Our model has produced an overall accuracy of 98.41%, an F1 score of 0.98, a precision of 0.98, and a recall of 0. 984. We tested the model with a total of 655 sample images (382 with pneumothorax and 273 without pneumothorax, as in Table 2) from the dataset without augmentation to ensure the classification accuracy. Out of 382 affected images, 375 images were correctly predicted as affected by the model and 7 images were wrongly predicted as not affected, when in reality they are affected. Similarly, in 273 not affected samples, 269 images were correctly predicted as not affected and 4 images were predicted as affected when they are actually not affected. An attempt was made to analyze the relation between disease affected and not affected. We observed that the SoftMax function used for binary recognition at the final stage of the model uses a time interval of 0:1 to adjoin the output available in the multiple neurons of the layer when the X-ray image is given as input to the PneumoNet model. Similarly, the probability of identifying the presence of the disease is 0.53, which is not affected by the pneumothorax probability at the output of the SoftMax function, which is 0.47.

Our proposed model uses the method of assessing the input image in three different dimensions. In other words, the model is trained with the obverse side of the X-ray image, followed by the flipside of the X-ray image. Finally, it congregates the values on both sides of the X-ray image and performs classification. This way of training results in an increased training loss, which effects the accuracy. Hence, in order to not compromise the accuracy of our model, the weight ratio between the three sides (obverse, flip, and both) of the image was adjusted to reduce the loss function. 

The loss function was adjusted to 1:0.6:0.6 for all three sides, respectively, resulting in a higher area under the curve (AUC).

This metric is used to describe the performance of the model. It is used to defy the disparity of the count in the input samples in a little amount of time. As per the values in Table 7, when the sequence of training is obverse–flip–both, the maximum accuracy is recorded as an average of 0.97. If the sequence is flip–obverse–both, the average accuracy is around 0.96. Similarly, when the sequence of training is performed on both sides, the average AUC value is recorded as 0.97. Correspondingly, the data presented in Table 8 express the ratio of weight of the proposed model tested at every moment of the input. Initially, the ratio is checked from 1 to 0.1 (on both sides), then the weight is gradually increased up to 1 to 0.6. The average ratio weight is stopped at 0.6 since the maximum accuracy level is reached and remains the same in other sequence ratios. Hence, only ratios up to 1:0.6:0.6 are recorded in Table 8.

For a visual representation of the proposed model output, we used the Grad-CAM [39] heat map method. The primary purpose of the heat map is to emphasize the significant portion of the chest X-rays that the model prioritizes highlighting. To aid in the target identification of chest X-ray images in detecting pneumothorax and the subsequent course of action, we employed the Grad-CAM approach that presents the output image in a colored representation. Any of the convolution layers can have Grad-CAM applied to it. Grad-CAM is typically applied to the final convolution operation in a network. The images in Figure 8 are very clear; thus, a radiologist can clearly observe color with Grad-CAM to help them work quickly and confidently. For an example of a heat map generated by the proposed model, see Figure 8, where the black patterned line indicates the affected region of the lungs.

We observe that when the ratio of weights between three sides if the X-ray image is adjusted, it results in an increased weighted average and macro value of the proposed PneumoNet model. 

The ratio of weight obtained at all three sides is shown in Table 9 and Figure 9. The receiver operating characteristic curves obtained for the entire model are shown in Figure 10. Notably, the outcome of the model is promising because data augmentation is conducted. If the model is trained with new positive samples, it leads to model overfitting, which is reflected as a reduced recognition rate and hence a reduced accuracy. The value of AUC supports the use of uneven input samples. Hence, the output obtained at various stages of our PneumoNet model is promising as it achieved 98.41% accuracy in recognizing the presence of a pneumothorax.

### 4.1. Analysis of Model’s Accuracy for Various Pneumothoraces

To evaluate the proposed model’s classification accuracy, we performed the analysis as described in the following. Undersized pneumothoraxes had up to 1 cm of visceral-parietal pleura separation in one lung area. A separation greater than 1 cm and less than 2 cm is classified as an average-sized pneumothorax. Similarly, separations greater than 2 cm were classified as a large pneumothorax [40]. We assessed our proposed approach with these three classifications with respect to the accuracy, specificity, and sensitivity of the model. The results are shown in Table 9. The results obtained show that the proposed model is highly effective in classifying outsized and moderate pneumothoraces, but its accuracy is reduced when recognizing undersized pneumothoraces. The specificity of the proposed approach is 97.2% for outsized pneumothoraces, with a sensitivity of 98.06% and an accuracy of 98.13%. Similarly, the sensitivity, specificity, and accuracy were 97.3%, 97.1%, and 98.1%, respectively, for average-sized pneumothoraces. Finally, we obtained a reduced specificity of 96.8%, a sensibility of 95.6%, and accuracy of 97.81% for undersized pneumothoraces. The other parameters, such as F1 score, recall, and precision, remain high for outsized and average pneumothoraces and lower for undersized pneumothoraces.

### 4.2. Ablation Study

Four experiments were performed as part of an ablation investigation by altering different parts of the proposed PneumoNet model. By altering different components, it is feasible to produce a more reliable design with an increased recognition rate. An ablation study was undertaken with the following variations: varying the dropout value, modifying both the dense and convolution layers, and adjusting the activation function and accuracy rate with and without the COT.

### 4.3. Ablation Study_1: Modifying the Number of Dense and Convolution Layers

We have presented the results obtained in Table 10 when the dense and convolution layers are modified. The table contains training accuracy, validation accuracy, and the inference obtained from the results. Seven convolution layers and five dense layers provided a training accuracy of 65.21% and a validation accuracy of 69.79%, which is the lowest obtained accuracy. Similarly, with six convolution layers [41] and four dense layers, the training and validation accuracies are 90.71% and 72.65%, respectively. When the numbers of convolution and dense layers are 5 and 4, the training and validation accuracies are 92.67% and 91.51%, and when the numbers of convolution and dense layers are 4 and 3, the training and validation accuracies are 93.87% and 93.88%, respectively. When the number of convolution layers is two and the number of dense layers is three, we achieved the highest training and validation accuracies of 98.4% and 98.3%, respectively.

### 4.4. Ablation Study_2: Adjusting the Activation Function

In Table 11, the data obtained while adjusting the activation function are shown. We used the ReLu, tanh, and Sigmoid activation functions to find the best result for training and validation [37]. When tanh was used, we achieved 88.92% and 90.62% as the training and validation accuracies, respectively. When the sigmoid activation function was used, the training accuracy was 91.32% and validation accuracy was 90.87%. Finally, we obtained the highest training and validation accuracies of 97.99% and 96.89%, respectively, when ReLu was used. Hence, ReLu was chosen as activation function for the proposed model.

### 4.5. Ablation Study_3: Varying the Dropout Value

Table 12 contains the results obtained for various dropout rates. When the dropout rate was 0.1, the model obtained a 98.39% training accuracy and a 98.36% validation accuracy, which is the highest accuracy. When the dropout rate was changed to 0.2, the validation and training accuracies were 94.72% and 91.75%, respectively. For a dropout rate of 0.15, the validation and training accuracies were 94.61% and 90.41%, respectively.

### 4.6. Ablation Study_4: With and without Channel Optimization

Table 13 contains the results obtained when the model was trained with and without the COT. When channel optimization is performed, the model produced the highest accuracy of 98.41% and an F1 score of 98.32%. On the other hand, when the model was trained without the COT, the accuracy and F1 score dropped to 94.68% and 93.98%, respectively.

By optimizing the channels, the issue of losing data due to differences in the significance of various channels in the feature maps during convolution pooling has been solved. The COT block enhanced the network’s ability to represent data by considering the connections between each channel and adjusting the features on a per channel basis. This enabled the network to learn how to selectively enhance the most relevant features while suppressing irrelevant ones using global information. Consequently, the network’s performance was improved.

## 5. Discussion

The objective of our proposed work is to provide a DL model to detect the presence of pneumothorax in X-ray images. Most of the existing methods are unable to extract the features of all the sides of the input X-ray image and the darker part of the image. Our proposed approach handles this issue using the COT to address the limitations found in several of the existing methods and by using hyperparameter optimization. Our accuracy results show the proposed model outperforms the existing models in detecting the disease, with the highest accuracy and minimum time. We compared our work with some existing DL approaches that use X-ray images for model training. The comparison results with other models are shown in Table 14. To verify the obtained results for their correctness, we also performed the test with different sized images of under and over 1 cm. The results obtained show that the proposed model is highly effective in classifying outsized and moderate pneumothoraces.

We found several limitations in our proposed approach. Pneumothorax can be difficult to spot on a chest X-ray image if there are overlapping structures present in the image. As a result, in these cases, the model cannot easily categorize the presence of the disease. Furthermore, the visual symptoms of a pneumothorax, which include a narrow line at the lung’s edges and a variation in the lung’s texture, might be challenging for our model to identify.

## 6. Conclusions

We propose a modest DL model, PneumoNet, to identify the existence of a pneumothorax in the human lungs. Although many methods using machine learning and DL models have been proposed, the results obtained from our proposed model outperform the existing approaches. If accuracy is used as the indicator to assess performance, then our model is the best approach. If the AUC, another measure to assess the model performance, is used, then our proposed model is also better at recognizing the presence of the disease. To assess our model, we use common machine learning and DL parameters, demonstrating the good performance of our proposed approach. We used the COT to improve the acknowledgment rate of the model. Data augmentation helps support the accurate augmentation of the images. If image augmentation is not conducted, the loss function increases. We used the affine augmentation tool to fine-tune the quality of the input X-ray images. Our model addresses the vanishing gradient issue, which is the most frequently observed problem in most existing approaches. The issue is solved with our COT method. Overall, our PneumoNet model can be effectively applied in the healthcare sector for recognizing the deadly disease of pneumothorax and can play a vital role in the field of the Internet of Medical Things.

## Figures and Tables

**Figure 1 diagnostics-13-01305-f001:**
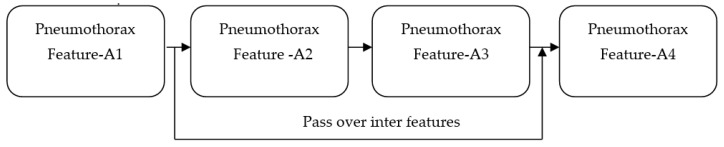
Addressing the vanishing gradient problem.

**Figure 2 diagnostics-13-01305-f002:**
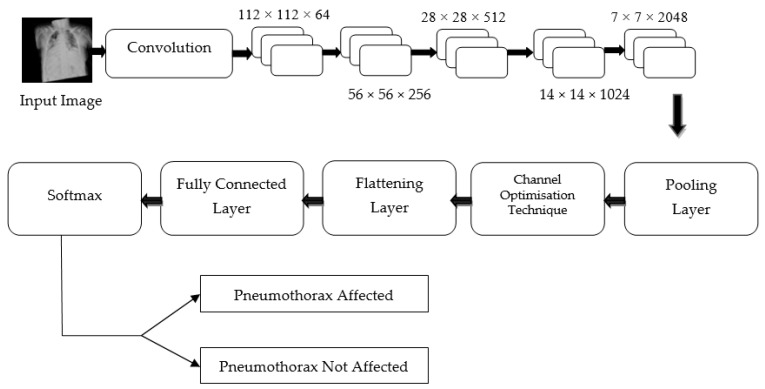
Proposed architecture.

**Figure 3 diagnostics-13-01305-f003:**
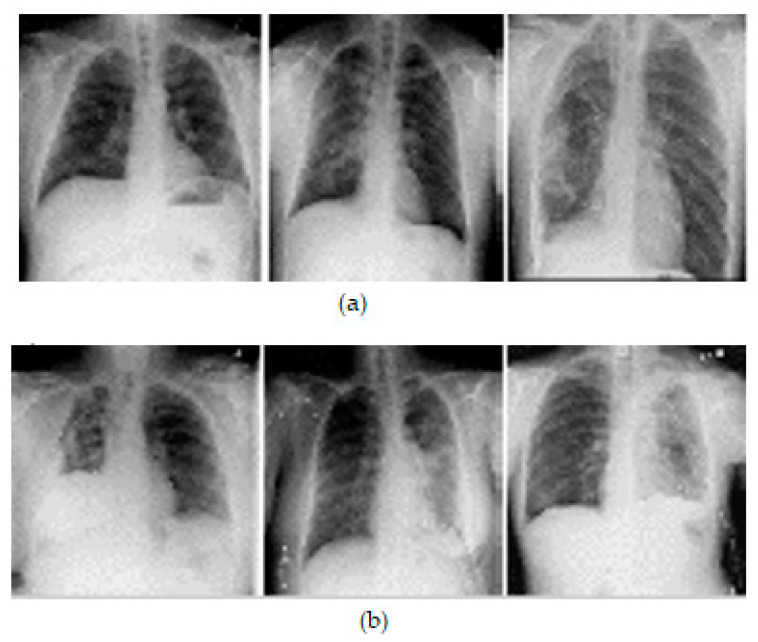
Samples of X-ray images: (**a**) pneumothorax affected and (**b**) pneumothorax not affected.

**Figure 4 diagnostics-13-01305-f004:**
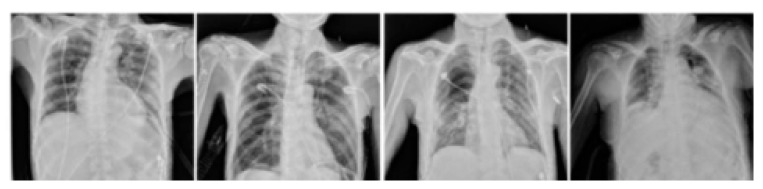
Preprocessed samples of chest X-ray images using the affine augmentation tool.

**Figure 5 diagnostics-13-01305-f005:**
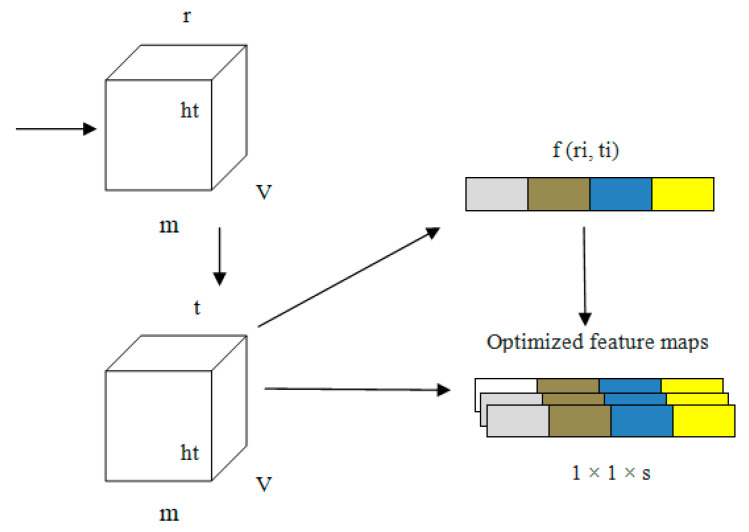
Structure of the channel optimizing technique (COT).

**Figure 6 diagnostics-13-01305-f006:**
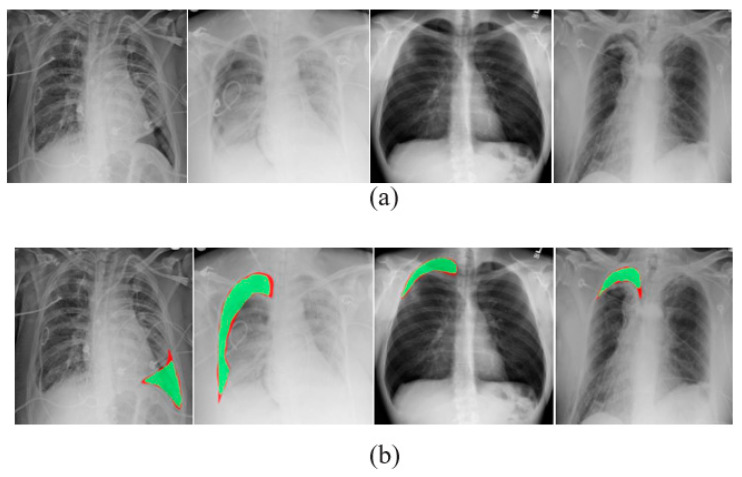
(**a**) Visually identified pneumothorax and (**b**) pneumothorax identified in X-ray images by PneumoNet.

**Figure 7 diagnostics-13-01305-f007:**
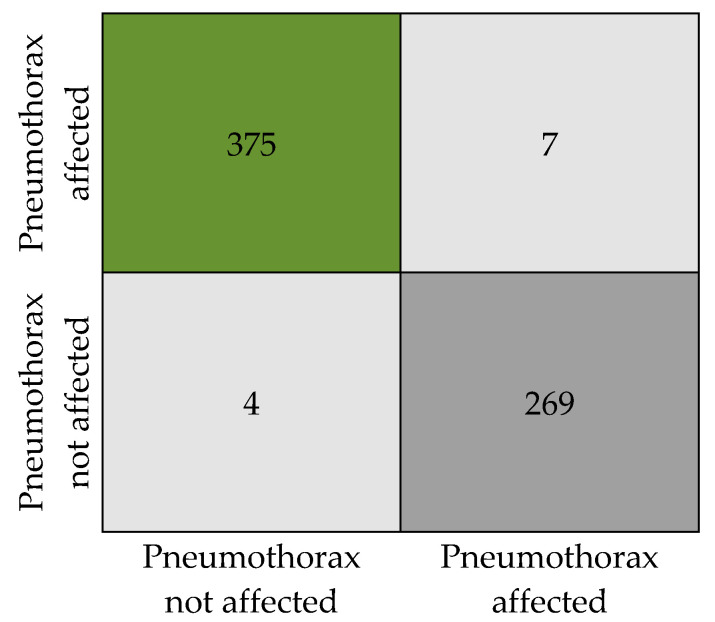
Confusion matrix.

**Figure 8 diagnostics-13-01305-f008:**
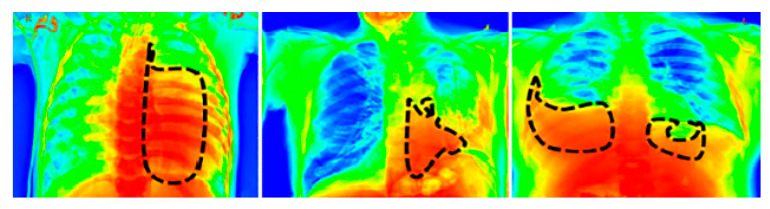
Grad-Cam heat map of the proposed model’s output.

**Figure 9 diagnostics-13-01305-f009:**
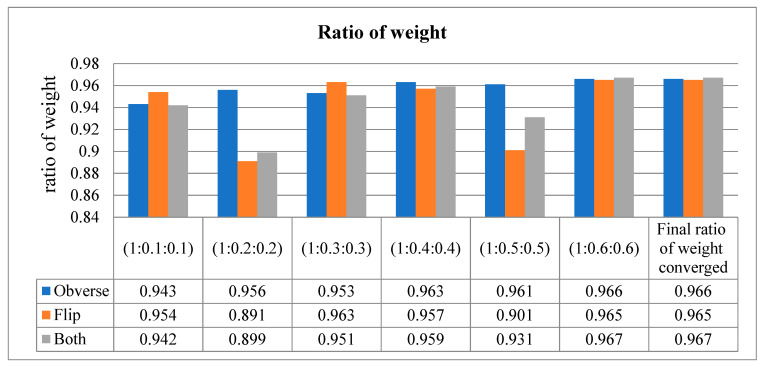
Ratio of weight (converged at 1:0.6:0.6).

**Figure 10 diagnostics-13-01305-f010:**
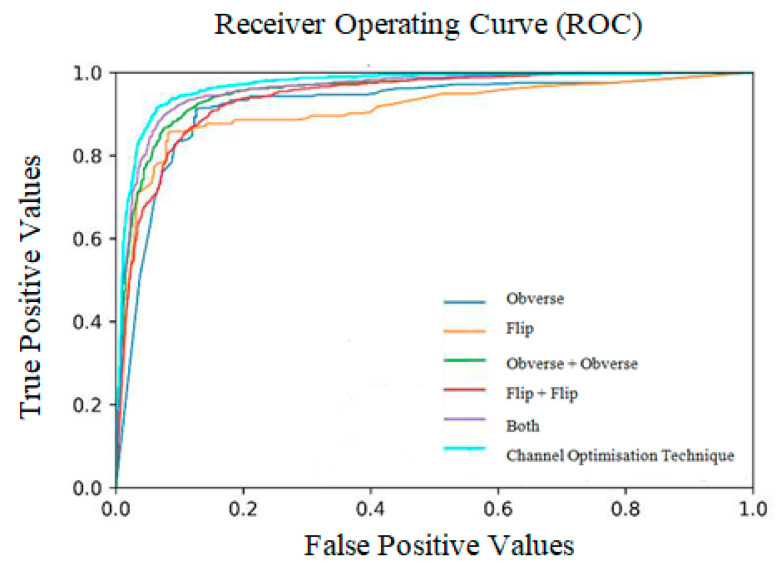
Receiver operating curves (ROC).

**Table 1 diagnostics-13-01305-t001:** Overview of related work.

Existing Methods	Methodology	Dataset	Samples Utilized	Accuracy
Greenspan et al. [31]	Texture analysis	Upright chest radiographs	10,832	87%
Pandiyan. et al. [37]	Ens4B-Unet	Challenge	12,047	86.06%
Sundaram et al. [24]	ResNet	SIIM	11,024	91.32%
R. Liao et al. [19]	Unet	SIIM	21,032	93.23%
S. Hamde et al. [34]	DCNN	Clinical imaging	30,000	94.38%
Riasatian et al. [35]	CNN	Self-collected	12,762	93.71%
S. Park et al. [36]	Patch-based	SIIM	14,201	92.81%
R. Yaakob et al. [25]	AlexNet	CXR dataset	15,372	93.74%

**Table 2 diagnostics-13-01305-t002:** Dataset parameters.

Parameters	Samples before Augmentation	Samples after Augmentation
Image resolution	1024 × 1024	124 × 124
Total no. of samples	3116	11,500
Classes	2	2
Samples trained with pneumothorax	1623	8300
Samples trained without pneumothorax	838	1100
Samples tested with pneumothorax	382	1300
Samples tested without pneumothorax	273	800

**Table 3 diagnostics-13-01305-t003:** Augmented chest x-ray images.

Image	No. of Images
Original X-ray image	120
Panned X-ray image	120
Spun X-ray image	120
Image rotated to 90°	120
Image rotated to 180°	120
Image rotated to 270°	120
Total images after one round of augmentation	720

**Table 4 diagnostics-13-01305-t004:** Results of hyperparameter optimization.

Hyperparameter Optimization	Metrics	Accuracy in %
Batch Size	8	83.87
	16	87.92
	32	95.98
	64	98.41
	128	96.82
	256	91.02
Learning rate	0.000461	86.31
	0.000372	91.09
	0.000201	95.82
	0.000173	98.41
	0.000199	97.81
	0.00021	96.12
Dropout rate in %	14.53	85.47
	5.72	91.11
	10.76	96.87
	11.02	98.41
	14.51	97.88
	15.78	96.17

**Table 5 diagnostics-13-01305-t005:** Computational complexities of the proposed approach.

Model	Parameters	Flops	Accuracy
Efficient Net	14.2 million	25.7 G	89.32%
MobileNet	10.14 million	24.81 G	96.75%
Inception Rennet	9.91 million	19.31 G	92.43%
Proposed PneumoNet	7.61 million	12.5 G	98.41%

**Table 6 diagnostics-13-01305-t006:** Overall results obtained from the proposed model.

Model Classification	Accuracy	F1 Score	Specificity	Recall	Precision
Pneumothorax Affected	98.41	0.983	0.978	0.981	0.981
Pneumothorax Not Affected	98.53	0.976	0.973	0.987	0.963

**Table 7 diagnostics-13-01305-t007:** Area under the curve values for all three sides of the input X-ray image.

Sequence of PneumoNet Training	Obverse	Flip	Both
Obverse–Flip–Both	0.959	0.961	0.973
Flip–Obverse–Both	0.962	0.971	0.966
Both	0.971	0.97	0.97
Overall AUC	0.961	0.963	0.974

**Table 8 diagnostics-13-01305-t008:** Ratio of weight obtained at all three sides of the input X-ray image.

Ratio of Weight (Both–Obverse–Flip)	Obverse	Flip	Both
(1:0.1:0.1)	0.943	0.954	0.942
(1:0.2:0.2)	0.956	0.891	0.899
(1:0.3:0.3)	0.953	0.963	0.951
(1:0.4:0.4)	0.963	0.957	0.959
(1:0.5:0.5)	0.961	0.901	0.931
(1:0.6:0.6)	0.966	0.965	0.967
Final ratio of converged weight	0.966	0.965	0.967

**Table 9 diagnostics-13-01305-t009:** Proposed method’s results for different sized pneumothoraces.

Model	Accuracy (%)	F1 Score (%)	Specificity (%)	Recall (%)	Precision (%)
Undersized pneumothorax (less than 1 cm)	97.81	96.40	96.64	93.47	95.91
Average pneumothorax (1 cm to 2 cm)	98.13	98.17	97.11	97.28	98.20
Outsized pneumothorax (more than 2 cm)	98.13	98.13	97.21	97.54	98.65

**Table 10 diagnostics-13-01305-t010:** Results of ablation study 1, modifying both the number of dense and convolution layers.

Set Up No.	Number of Convolution Layer	Number of Dense Layer	Training Accuracy	Validation Accuracy	Inference
1	7	5	65.21%	69.79%	Accuracy is low
2	6	4	90.71%	72.65%	Accuracy is low
3	5	4	92.67%	91.51%	Accuracy is low
4	4	3	93.87%	93.88%	Accuracy is low
5	3	3	95.67%	94.81%	Accuracy is low
6	2	2	98.40%	98.31%	Highest Accuracy

**Table 11 diagnostics-13-01305-t011:** Results of ablation study 2, adjusting the activation function.

Set Up No.	Activation Function Used	Training Accuracy	Validation Accuracy	Inference
1	Sigmoid	91.32%	90.87%.	Accuracy is low
2	Tanh	88.92%	90.62%	Accuracy is low
3	ReLu	97.99%	96.89%	Highest Accuracy

**Table 12 diagnostics-13-01305-t012:** Results of ablation study 3, varying the dropout value.

Set Up No.	Dropout Rate	Training Accuracy	Validation Accuracy	Inference
1	0.2	94.72%	91.75%	Accuracy is low
2	0.15	94.61%	90.62%	Accuracy is low
3	0.1	98.39%	98.36%	Highest Accuracy

**Table 13 diagnostics-13-01305-t013:** Results of ablation study 4, models without channel optimization.

Set Up No.	Presence of COT	Accuracy	F1 Score	Inference
1	Without COT	94.68%	93.98%	Accuracy and F1 score are low
2	With COT	98.41%	98.32%	Highest accuracy and F1 score

**Table 14 diagnostics-13-01305-t014:** Performance comparison of existing and proposed methods.

Model	Dataset Used	Accuracy (%)	F1 Score (%)	Specificity (%)	Recall (%)	Precision (%)
Efficient Net [5]	SIIM	89.32	90.3	84.6	93.4	87.6
Inception Rennet [9]		92.43	89.8	89.6	89.4	91.3
Xception [16]		84.15	91.3	85.8	89.2	91.7
MobileNet [28]		96.75	93.4	92.3	89.9	94.8
ResNet [35]		97.68	94.6	97.2	92.4	93.6
Proposed PneumoNet		98.4	98.3	97.8	98.5	98.1

## Data Availability

The dataset utilized in this investigation is publicly available and can be downloaded from https://kaggle.com/competitions/siim-acr-pneumothorax-segmentation, accessed on 10 February 2022.

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
