# Peer review of "“Quo Vadis Diagnosis”: Application of Informatics in Early Detection of Pneumothorax"

_diagnostics, 2023, doi:10.3390/diagnostics13071305_

Round 1

Reviewer 1 Report

Dear Editor and Authors,

Thank you for asking me to review this manuscript titled “Quo vadis diagnosis”: application of informatics in early detection of pneumothorax by Dr. Kumar and colleagues.

In this study the authors utilize a machine learning/artificial intelligence program to screen and elucidate the ability to identify pneumothoraxes on chest X-ray! They report quite encouraging results following their protocol. This is a well reported study, interesting but needs some language correction so that it becomes more clear (please pay attention especially with the tense)

Otherwise it is well presented with nice graphs and methodology.  I do have a few comments (please see below:)

COMMENTS:

1. The abstract is quite disjointed and unstructured, also it reads as an introduction and has no focus. The results are presented only briefly at the end. There is no conclusion. Basically all of the abstract needs re-writing.

2. What is “clear reappearance”? Explain what you mean.

3. “Chest X-rays are commonly utilized to diagnose and follow-up numerous disorders, such as hepatocellular carcinoma, tuberculosis (TB), asthma, and lung damage” this phrase does not make sense at al, please re-phrase/edit.

4. “If air is present even in low quantity in the laceration area” what is the laceration area? Please explain/correct!!

5. You don’t detect something manually (mana=hands) you detect it visually!! Correct throughout the text!

6. The material and methods section needs re-writing in the past tense. You did all this things you do not plan to perform them now!

7. “The process is continued until the dataset contains 3000 image samples” How did you reach to this number? Was there a sample size calculation performed to demonstrate the adequate sample to use?

I thought the original images were only 120 and you augmented/enhanced them as part of the testing approach.

8. The results look quite good = 98.41% detection!!

9. You do not present results in the discussion. Please use the results section for that and edit the discussion.

In conclusion, this is an interesting study but it needs some major editing and cleaning up before it is ready to be presented in the clinical community. Nothing which can’t be changed as methodologically the work is sound. Thank you again and wishing well to all. Awaiting your revision!

Author Response

Respected Reviewer,

We are writing to express our sincere gratitude for the time and effort you took to provide feedback on our article, “Quo vadis diagnosis”: application of informatics in early detection of pneumothorax”. Your positive comments were greatly appreciated and have given us a renewed sense of confidence in our work.

In particular, we found your comments on the clarity of our writing and the strength of our arguments to be particularly helpful and encouraging. It is always gratifying to receive feedback from a respected colleague in the field.

Your feedback has motivated us to continue working on our research and to strive for excellence in our writing. It is through constructive criticism and feedback such as yours that we are able to improve our skills and contribute meaningfully to the field.

We have addressed all the comments provided by the reviewer and have updated the same in the manuscript for the reviewer’s kind consideration.

Once again, we would like to express our gratitude for your thoughtful review of our article.

Sincerely,

Oana GEMAN

Associate Professor, Ph.D., Habill.

Medical Bioengineer

IEEE Senior Member

Stefan cel Mare University of Suceava, Romania
 [email protected]

Maria Daniela Craciun

 Lecturer Ph. D

Interdisciplinary Research Centre in Motricity Sciences and Human Health,

 Åžtefan cel Mare University of Suceava, 720229

 Suceava,

 Romania,

[email protected]

Reviewer comments and authors response

Dear Editor and Authors,

Thank you for asking me to review this manuscript titled “Quo vadis diagnosis”: application of informatics in early detection of pneumothorax by Dr. Kumar and colleagues.

In this study the authors utilize a machine learning/artificial intelligence program to screen and elucidate the ability to identify pneumothoraxes on chest X-ray! They report quite encouraging results following their protocol. This is a well reported study, interesting but needs some language correction so that it becomes more clear (please pay attention especially with the tense)

Otherwise it is well presented with nice graphs and methodology.  I do have a few comments (please see below :)

COMMENTS:

  1. The abstract is quite disjointed and unstructured; also it reads as an introduction and has no focus. The results are presented only briefly at the end. There is no conclusion. Basically all of the abstract needs re-writing.

Thank you for your feedback regarding the abstract. We apologize for the lack of structure and coherence in the current version. In response to your comment that the abstract reads like an introduction and has no focus, we ensure that the abstract is revised and believe that the abstract clearly defines the research question, highlights the main findings, and provides a clear conclusion. We have also presented the results more prominently to address your feedback that they were not given sufficient attention in the original version.

The rewritten abstract is updated in the manuscript for the reviewer’s kind consideration.

  1. What is “clear reappearance”? Explain what you mean.

We apologize for inconsistent term, instead of mentioning “clear visual representation “it’s been represented as “clear reappearance”. The error is corrected as per the reviewer’s guidance and updated in Section 2, Page No2., paragraph 3.

  1. “Chest X-rays are commonly utilized to diagnose and follow-up numerous disorders, such as hepatocellular carcinoma, tuberculosis (TB), asthma, and lung damage” this phrase does not make sense at all, please re-phrase/edit.

As per the reviewer’s instructions, the lines are rephrased.

We tried to brief about other lung related disease that could be visualized through X-ray images.  We apologize for any confusion caused in our previous text. We acknowledge that our attempt to provide a brief information about other medical conditions that could be visualized through X-ray images may have deviated from the intended focus of the text. The corrected phrase could be found in Section 2, Page.No2, paragraph-3.

  1. “If air is present even in low quantity in the laceration area” what is the laceration area? Please explain/correct!!

Actually laceration area is a minor tear in the lung tissue due to injury. We apologize for not explaining this earlier. As per the reviewer's guidance, we have corrected the mistake in the manuscript, which can be found in Section .2, page number 2, paragraph-4.

  1. You don’t detect something manually (mana=hands) you detect it visually!! Correct throughout the text!

We regret and thank the reviewer for pointing out the mistake. All “manual, manually” are replaced in the entire document as per the reviewer’s feedback. The changes are highlighted for kind consideration.

  1. The material and methods section needs re-writing in the past tense. You did all this things you do not plan to perform them now!

Thank you for bringing this to our attention. It is important to maintain consistency in tense throughout the section to avoid confusion and ensure clarity. We apologize for any confusion our previous response may have caused and we will ensure to use the past tense appropriately in the future. The methods and materials section is updated with past tense as per the reviewer guidance and could be found in Section 3 from page no.6 to 14.

  1. “The process is continued until the dataset contains 3000 image samples” How did you reach to this number? Was there a sample size calculation performed to demonstrate the adequate sample to use?I thought the original images were only 120 and you augmented/enhanced them as part of the testing approach.

"Thank you for your question. We agree that it is important to have a justification for the sample size used in the dataset. In this study, we opted to use a total of 3000 image samples in order to provide a sufficient number of samples for the proposed PneumoNet to learn from. We did not perform a formal sample size calculation, but rather chose this number based on our experience with similar image classification tasks and considering the computational resources available.

Regarding your observation about the original images, you are correct that we started with a set of 120 images. However, we used data augmentation techniques to generate additional samples by applying random transformations to the original images. This allowed us to increase the diversity of the dataset and reduce the risk of overfitting. In total, we generated 3000 image samples from the original set of 120 images through data augmentation. We hope this explanation clarifies how we arrived at the sample size used in our study."

  1. The results look quite good = 98.41% detection!!

We express our sincere thanks to the reviewer for the valuable appreciation and motivation.

  1. You do not present results in the discussion. Please use the results section for that and edit the discussion.

As per the reviewer’s guidance the results are presented in the results section and the updated details could be found in Section.4, from page no 14 to 21 of the manuscript.

Reviewer 2 Report

This paper presents a chest image classification using a customized CNN-based classification approach.

1. It states in the abstract that the paper addresses two limitations, including class disparity and detecting the dark portion in the X-ray images. However, Figure 1 states that the challenge is the vanishing gradient issue. This is not consistent.
2. It states in the abstract that the segmentation method is proposed to detect the dark portion. However, it is not described in the main text and not evaluated in experiments. 
3. The paper structure should be revised. The proposed method (Section 2.e) should be moved earlier. All experimental results should be moved to Section 3, which starts with the overall performance evaluation, followed by various ablation studies.
4. The channel optimization technique is introduced in Figure 5. What is "optimized" here? Is this a trainable layer? If so, have any additional trainable parameters been introduced? There is no ablation study on this point.
5. Why do the input images need to be resized to a very smaller size than the original images in Table II?
6. Are all methods presented in Table 1 using the same dataset? If not, there is no point in comparing their accuracy performance because different datasets were used in their studies.
7. Figure 2 is displayed wrongly; its caption is partially occluded.

Author Response

Respected Reviewer,

We would like to express my sincere gratitude for taking the time to review our paper entitled “Quo vadis diagnosis: application of informatics in early detection of pneumothorax”. Your insightful comments and valuable suggestions have helped us to improve the quality of our research significantly.

Your expertise and attention to detail have been invaluable in identifying areas that required further clarification and enhancement. Your suggestions for improving the methodology and analysis have been particularly helpful, and we have implemented them to strengthen the paper.

Once again, we thank you for your time and effort in reviewing our paper. Your feedback has been instrumental in shaping the final product, and we are grateful for your contribution to this work.

We have presented brief explanation for the comments and the same have been updated in the article for your kind consideration.

Sincerely,

Oana GEMAN

Associate Professor, Ph.D., Habill.

Medical Bioengineer

IEEE Senior Member

Stefan cel Mare University of Suceava, Romania
 [email protected]

Maria Daniela Craciun

Lecturer Ph.D

Interdisciplinary Research Centre in Motricity Sciences and Human Health,

 Åžtefan cel Mare University of Suceava, 720229

 Romania,

[email protected]

Reviewer comments and authors responses:

This paper presents a chest image classification using a customized CNN-based classification approach.

  1. It states in the abstract that the paper addresses two limitations, including class disparity and detecting the dark portion in the X-ray images. However, Figure 1 states that the challenge is the vanishing gradient issue. This is not consistent.

In response to the reviewer's comment, it appears that there was an error in the abstract section of the manuscript regarding the limitations addressed by the proposed PneumoNet model. While Figure 1 does indeed show the vanishing gradient issue as a challenge in detecting X-ray images, we admit our mistake and the abstract must have accurately reflected the limitations addressed in the paper.

To clarify, the proposed PneumoNet model addresses the limitations of class disparity and detecting the dark portions in X-ray images. While the vanishing gradient issue is also a challenge that needs to be addressed in PneumoNet model, it is not one of the specific limitations addressed by the proposed model in this paper.

We apologize for any confusion caused by this inconsistency and ensure that the abstract is revised to accurately reflect the limitations addressed by the proposed PneumoNet model. The revised abstract is presented for reviewer’s kind consideration and the same have been updated in Page.No.1 of the manuscript.

  1. It states in the abstract that the segmentation method is proposed to detect the dark portion. However, it is not described in the main text and not evaluated in experiments. 

We appreciate the reviewer's feedback and apologize for any confusion caused by the discrepancy between the abstract and the main text. The segmentation method proposed in the abstract was intended to detect the dark portion, but due to time constraints and limited resources, we were unable to fully describe and evaluate this method in the main text. We apologize for any oversight on our part and the manuscript is updated in Section 3 Page No.10, Paragraph 2 with the segmentation process done in our experiment.

Table XI contains the results of existing approaches that had not used any segmentation process. We regret for not mentioning that the existing approaches had not used segmentation in an emphasized manner. We appreciate the reviewer's input in bringing this to our attention.

  1. The paper structure should be revised. The proposed method (Section 2.e) should be moved earlier. All experimental results should be moved to Section 3, which starts with the overall performance evaluation, followed by various ablation studies.

We thank for the reviewer's feedback and agree that the paper structure could be revised for better clarity and organization. We apologize for any confusion caused by the current structure and have revised the paper accordingly.

We agree that the proposed method should be moved to an earlier section for better flow and organization. In addition, we will move all experimental results to Results Section, as suggested by the reviewer. Section 4 will start with the overall performance evaluation, followed by various ablation studies. This should provide a clear and concise presentation of our approach and results. (Page No. 14 to 21)

  1. The channel optimization technique is introduced in Figure 5. What is "optimized" here? Is this a trainable layer? If so, have any additional trainable parameters been introduced? There is no ablation study on this point.

We again regret for the inconsistent information provided about Channel Optimization Technique (COT). A brief explanation is provided for the consideration of the reviewer. The manuscript is updated with ablation study 4 of COT in Table X (d), in Section 4, Page No.21.

Channel optimization is a block, added to the proposed PneumoNet model. This block contains a set of trainable layers that can be inserted into our proposed neural network architecture to improve their performance.

The main reason for COT is, it is built with the trainable parameters to perform the following operation. The first layer in COT reduces the spatial dimensions of the input feature maps and produces a channel descriptor. The channel descriptor is used to selectively weight the feature maps. Therefore, the COT block is composed of trainable layers and can learn the optimal weighting of the feature maps based on the input.

After the Channel optimization the issue of loss caused by the varying importance of different channels of feature maps in the convolution pooling process was resolved. COT block enhances the network's ability to represent data by modeling the dependence between each channel and adjusts the features channel by channel. This allows the network to learn to selectively strengthen the features that are most relevant while suppressing irrelevant features through global information.

An ablation study was conducted to investigate the impact of using COT on the performance of pneumothorax detection using chest x-ray images.

The results of the study showed that incorporating COT into the PneumoNet model significantly improved the performance of pneumothorax detection. The model with COT achieved an accuracy of 98.41% and an F1-score of 98.32%, while the model without COT achieved an accuracy of 94.68%% and an F1-score of 93.98%.

The study also revealed that the COT helped the model to learn to selectively strengthen informative features while suppressing irrelevant features, resulting in more accurate and reliable predictions. These findings demonstrate the effectiveness of PneumoNet in improving the performance of pneumothorax detection using chest x-ray images.

  1. Why do the input images need to be resized to a very smaller size than the original images in Table II?

Thank you for the reviewer’s comment. The main reasons behind resizing the images to a smaller size are,

Computational efficiency: Resizing the input images to a smaller size reduces the number of pixels that the model needs to process, which can significantly reduce the computational cost of training and testing the model.

Memory constraints: In addition to computational efficiency, resizing the images can also help to conserve memory resources. Large images require a significant amount of memory to store and process, which can be a challenge for systems with limited memory capacity.

Feature extraction: Resizing the images can help to enhance the features in the image that are relevant to the task at hand. In the case of pneumothorax detection, resizing the images can help to highlight the areas of the image that are most important for identifying the condition where the air leaks, and dark region in the x-ray images .

  1. Are all methods presented in Table 1 using the same dataset? If not, there is no point in comparing their accuracy performance because different datasets were used in their studies.

We wish to bring to the kind notice of the reviewer that the methods presented in Table 1used SIIM dataset for the detection of Pneumothorax.

  1. Figure 2 is displayed wrongly; its caption is partially occluded.

We regret for the inconsistent figure. The mistake is corrected as per the reviewer’s instruction.

Round 2

Reviewer 1 Report

Dear Editor and Authors,

I have re-evaluated the revised manuscript. It is improved although it will still benefit from some language editing. The authors have answered my comments to a satisfactory level. I think we can now accept the work.

Thank you once more time for asking me to review this work  and good luck to all!

Kind regards,

Reviewer 2 Report

The revision is fine, there is no further comments.